# Current evidences of the efficacy of mosquito mass-trapping interventions to reduce *Aedes aegypti* and *Aedes albopictus* populations and *Aedes*-borne virus transmission

**Ali Jaffal**[1], **Johanna Fite**[1☯*], **Thierry Baldet**[2☯], **Pascal Delaunay**[3☯], **Frédéric Jourdain**[4,5☯], **Ronald Mora-Castillo**[6☯], **Marie-Marie Olive**[2,5☯], **David Roiz**[5]

**1** French Agency for Food, Environmental and Occupational Health & Safety (Anses), Maisons-Alfort, France, **2** ASTRE (Animal, Santé, Territoires, Risques, Ecosystèmes), CIRAD, Univ Montpellier, Montpellier, France, **3** Service de Parasitologie-Mycologie, Hôpital L'Archet, Centre Hospitalier Universitaire de Nice, UCA, Nice, France, **4** Santé publique France (French National Public Health Agency), Montpellier, France, **5** MIVEGEC, IRD, CNRS, Université Montpellier, Montpellier, France, **6** World organisation for animal health, Paris, France

☯ These authors contributed equally to this work.
* Johanna.fite@anses.fr

**Data Availability Statement:** All relevant data are within the manuscript and its Supporting Information files.

## Abstract

### Background

Over the past decades, several viral diseases transmitted by *Aedes* mosquitoes—dengue, chikungunya, Zika—have spread outside of tropical areas. To limit the transmission of these viruses and preserve human health, the use of mosquito traps has been developed as a complement or alternative to other vector control techniques. The objective of this work was to perform a systematic review of the existing scientific literature to assess the efficacy of interventions based on adult mosquito trap to control *Aedes* population densities and the diseases they transmit worldwide.

### Methods and findings

Following the Preferred Reporting Items for Systematic Reviews and Meta-Analyses (PRISMA) guidelines, a systematic review was conducted using the PubMed and Scopus databases. Among the 19 selected papers, lethal ovitraps were used in 16 studies, host-seeking female traps in 3 studies. Furthermore, 16 studies focused on the control of *Ae. aegypti*. Our review showed great heterogeneity in the indicators used to assess trap efficacy: e.g., the number of host-seeking females, the number of gravid females, the proportion of positive containers, the viral infection rate in female mosquitoes or serological studies in residents. Regardless of the type of studied traps, the results of various studies support the efficacy of mass trapping in combination with classical integrated vector control in reducing *Aedes* density. More studies with standardized methodology, and indicators are urgently needed to provide more accurate estimates of their efficacy.

**Funding:** The author(s) received no specific funding for this work.

**Competing interests:** The authors have declared that no competing interests exist.

## Conclusions

This review highlights gaps in the demonstration of the efficacy of mass trapping of mosquitoes in reducing viral transmission and disease. Thus, further large-scale cluster randomized controlled trials conducted in endemic areas and including epidemiological outcomes are needed to establish scientific evidence for the reduction of viral transmission risk by mass trapping targeting gravid and/or host-seeking female mosquitoes.

### Author summary

Over the past decades, several viral diseases transmitted by *Aedes* mosquitoes—dengue, chikungunya, Zika—have spread outside of tropical areas. To limit the transmission of these viruses and preserve the environment and human health, the use of mosquito traps has been developed as a complement or alternative to other vector control techniques.

This review supports the efficacy of mass trapping in combination with classical integrated vector control on the reduction in mosquito vector populations within several weeks of deployment. However, this research highlights gaps in the demonstration of the efficacy of mass trapping of mosquitoes in reducing viral transmission and disease. Thus, further studies are needed to establish the scientific evidence for the reduction of viral transmission risk by mass trapping targeting gravid and/or host-seeking female mosquitoes.

## Introduction

Over the past 50 years, the world has seen the emergence and dramatic spread of *Aedes*-borne arboviral diseases such as dengue, chikungunya and Zika, which are transmitted by the two invasive mosquitoes *Aedes aegypti* and *Ae. albopictus* [1]. Diseases caused by these viruses are most prevalent in tropical regions transmitted principally by *Ae. aegypti* and their incidence are increasing in temperate regions of the world, mainly due to the geographical expansion of *Ae. albopictus*. Several factors contribute to this risk: globalization of trade and travel, unplanned urbanization, climatic changes, water storage and waste disposal and limitations in the effectiveness of disease or mosquito control strategies [2]. In the last two decades, the number of dengue cases reported to the World Health Organization (WHO) has increased worldwide by eight folds [3], and epidemics of chikungunya and Zika have emerged outside their original distribution area [1]. Therefore, the control of *Aedes*-borne diseases becomes a public health and socio-economic burden [4]. In the absence of a vaccine and specific curative treatments, the only way to reduce the transmission of these diseases is to control mosquito vector populations or to reduce human-vector contact. To achieve this, a variety of control methods can be implemented (e.g., chemical, biological, genetic, mechanical control, trapping) [5]. For effective vector control, an integrated, proactive and evidence-based vector control strategy should be developed, which should be an optimal combination of several tools and techniques adapted to the local scenario and to the available resources [6]. Despite the health, economic and social importance of *Aedes*-borne diseases and the existence of numerous studies evaluating vector control methods and strategies, the evidence for the public health value of *Aedes* vector control is unfortunately weak [6].

Larval control methods, and in particular, community-based source reduction and larviciding, is a dominant paradigm for *Aedes* mosquito control [6]. These methods targeting larvae

show moderate efficacy in operational conditions but to maximize their impact on the reduction of virus transmission is important to reduce *Aedes* density and longevity [6]. Although the use of insecticides remains the predominant strategy for controlling adult female *Aedes* in case of epidemics, there is limited entomological and epidemiological evidence on the efficacy of ultralow volume (ULV) space spraying with some evidence of efficacy in some settings for Indoor Residual Spraying (IRS) or insecticide-impregnated materials [6,7,8,9,10,11]. It is important to note that authorities and citizens worldwide have a growing aversion to the use of insecticides, due to their limitations in efficacy, the emergence of insecticide resistance and their environmental and health impacts. Therefore, the development and evaluation of non-insecticidal tools targeting different fractions of the target population (host-seeking females, blood-fed females, etc.) should be fostered. Significant progress has been made and new techniques such as the Sterile Insect Technique (SIT), Incompatible Insect Technique (IIT), genetically modified mosquitoes, trapping, and spatial repellents are currently in different stages of development and have already provided the scientific community with an array of data on their efficacy [5]. Recently, there is entomological evidence for the mass deployment of lethal oviposition traps to reduce *Aedes* [mosquito density]. Our work complements and updates fhe review by Johnson *et al.* [12] and differs from it in two aspects. Firstly, from a methodological point of view, we have chosen a systematic review, which facilitates the updating of evidence as new results are available. Secondly, we have broadened the scope of mass trapping from gravid traps to host-seeking traps, with a view to considering several tools that can be included in an integrated pest management strategy. Mass trapping can be a low-cost, community-based, and sustainable engagement approach, attractive to complement other tools that can be selected locally within an integrated *Aedes* management strategy. There is no silver bullet for *Aedes*-eases control, and the most practical and productive path forward is to strengthen the evidence base so that, where appropriate, an effective "toolbox" can be deployed in an integrated manner taking into account the local situation and available resources [6].

There is a wide variety of commercially available traps targeting the different life stages of a mosquito (eggs, larvae, pupae or adults). Historically, these traps were used to monitor the distribution, abundance and infection rate of mosquito populations [13]. Next, the traps were used to evaluate their efficacy as a vector control method [14]. Indeed, over the past decade, some of these traps have been evaluated as vector control tools, mainly targeting adult females [12]. Depending on the physiological stage targeted, two categories can be distinguished: (i) host-seeking female traps and (ii) lethal ovitraps targeting gravid females. Other stages could potentially be targeted (adults seeking a sugar meal, or adults swarming during the mating period) [15]. Females of *Ae. albopictus* and *Ae. aegypti* are hematophagous and have olfactory receptors that allow them to detect odor and carbon dioxide ($CO_2$) produced by their hosts. Traps targeting host-seeking females (e.g., BG-Sentinel, BG-Mosquitaire (Biogents, Regensburg, Germany), Mosquito Magnet (American Biophysics Corporation)) operate on the same principle: attracting mosquitoes by releasing carbon dioxide and/or attractants, either visual or olfactory (e.g., simulating the human body odor such as lactic acid or oct-1-en-3-ol) [16]. In all cases, the attracted mosquitoes are suctioned up by electric ventilation and caught into a collection bag. Lethal ovitraps (e.g., Centers for Disease Control and Prevention-Autocidal Gravid Ovitrap (CDC-AGO); Biogents Gravid Aedes Trap (BG-GAT); Sticky Ovitrap (SO), etc.) simulate a breeding site and use stagnant water, to which an infusion of plant material can be added, to attract gravid females of *Aedes* mosquitoes. Ovitrap can be lethal to females by a variety of means (insecticide-treated strips, sticky strips, etc.) [11]. Females of *Ae. albopictus* and *Ae. aegypti* have a specific behavior of laying eggs in man-made containers, especially in urban environments. Several studies have shown an association between urbanization and the distribution and density of these both *Aedes* vectors [17]. To clarify the place of mass trapping in the

overall strategy for the control of *Aedes*-borne diseases, there is a need to review evidence on the efficacy of mosquito traps in controlling the vector population, and reducing arboviruses transmission according to epidemiological contexts. Therefore, the objective of this work was to perform a scoping review of the existing scientific literature to assess the efficacy of mosquito trap-based interventions in controlling *Aedes* population density and the diseases they transmit, informing control intervention as well as building proof of concept to guide future studies.

## Methods

### Research question

A Population, Intervention, Comparator and Outcome (PICO) statement (Table 1) was developed to answer the following question: Are mosquito trap-based interventions effective in reducing *Aedes* populations and transmission of *Aedes*-borne diseases?

The scoping review focused on adult mosquito (gravid and host-seeking females) traps used to control *Aedes* populations. Thus, traps used exclusively for surveillance were not considered.

### Search strategy

The systematic and simultaneous searches were conducted on 25 February 2021 in two major databases (Scopus, a multidisciplinary database and PubMed, a primary life sciences database) with no limitations on the year of publication. The following search strings were used to identify references relevant to the research question:

Searching string used in PubMed:(("Aedes"[Title/Abstract]) AND (*trap*[Title/Abstract]) AND (control*[Title/Abstract] OR remov*[Title/Abstract] OR suppress*[Title/Abstract] OR eliminat*[Title/Abstract] OR reduc*[Title/Abstract]))

Searching string used in Scopus:

(Aedes) AND (*trap*) AND (control* OR remov* OR suppress* OR eliminat* OR reduc*)

The whole procedure of the review followed the recommendations from Preferred Reporting Items for Systematic Reviews and Meta-Analyses (PRISMA) Statements [18].

### Eligibility criteria

The inclusion and exclusion criteria for the review are shown in Table 2.

### Study selection

Articles were evaluated for inclusion, according to the eligibility criteria. The selection of references for inclusion was made using the CADIMA tool (CADIMA is a free web tool facilitating the conduct and assuring for the documentation of systematic reviews, systematic maps and further literature reviews; https://www.cadima.info/) with a first step based on a review of the title and abstract (considering the first three inclusion criteria). When it was unclear whether

**Table 1. Definition of PICO statement.**

|  | Definition |
|---|---|
| **Population** | *Aedes albopictus or Aedes aegypti* |
| **Intervention** | use of traps to control mosquito populations |
| **Comparator** | Intervention site compared to a control site or baseline data obtained at the same site |
| **Outcomes** | entomological and/or epidemiological and/or sociological indicators |

**Table 2. Inclusion and exclusion criteria.**

|  | Inclusion criteria | Exclusion criteria |
|---|---|---|
| **Date / Peer-review** | Peer-reviewed articles published before February, 25 2021 | Non-peer-reviewed articles or articles published after 25 February 2021 |
| **Population** | *Aedes albopictus* or *Aedes aegypti* | Other mosquitoes (other *Aedes*, *Culex*, *Anopheles*...) or insect species |
| **Intervention** | Use of lethal ovitraps or host-seeking female traps to control mosquito populations | No trap or other types of traps |
| **Comparator** | Intervention site compared to a control site or baseline data obtained at the same site | No control neither baseline data |
| **Outcomes** | Quantified entomological and/or epidemiological and sociological indicators | No indicators to quantify the efficacy of the intervention |
| **Study type** | Analytical studies (experimental or observational) that quantify the effect of an outcome | Non experimental or observational studies (reviews, laboratory studies...) |
| **Language** | Articles written in English, Spanish or French. | Languages other than English, Spanish or French. |

an article met the inclusion criteria based on the title and abstract, or when the reviewers disagreed, the full text of the manuscript was examined. Then, a second step based on full-text review (taking into account all inclusion criteria) was performed. At each selection stage, two reviewers evaluated each reference in a double-blind manner and any disagreement was resolved by discussion or consensus.

## Data extraction and analysis

The following information was extracted from studies included in the review: author names, years of publication, epidemiological context, target species, geographic location, experimental protocol, duration of the study, period of the year, traps used, surface of the intervention area, number of traps, coverage (% of houses treated), indicators used to measure trapping efficacy, and study conclusions. The analytical study designs were classified according to the typology proposed by the WHO (with experimental and observational categories) [15]. Reviewer completed the data for each study, one extraction using a template specifying the relevant data fields. All authors discussed the quality of the included studies. A spreadsheet was completed to collect and evaluate information on study design, coverage area, local environment and duration of the interventions in order to assess the quality of the selected studies and their alignment with the objectives of our analysis.

## Results and analysis of included studies

### Article searches and screening

The systematic search identified 2,033 articles describing studies using adult mosquito traps, as presented in the PRISMA flow chart (Fig 1). These included 2,032 articles from PubMed and Scopus databases and 1 article identified by screening cross-references. Of the 2,032 citations, 639 were duplicates, 1,134 articles were title and abstract checked, and 239 were full-text checked for eligibility (Fig 1). The most common reason for exclusion was the lack of a comparator (control or baseline data). Many articles were excluded for more than one reason. The "sociological indicators" in the selection criteria, intended to select studies of mass trapping against *Aedes* that included entomological and sociological indicators. However, the two studies with sociological indicators do not have entomological indicators [19,20]. At least, 19 articles fulfilled the inclusion one of the entomological efficacy studies of mass trapping that we retained included sociological indicators. criteria, according to the PICO statement.

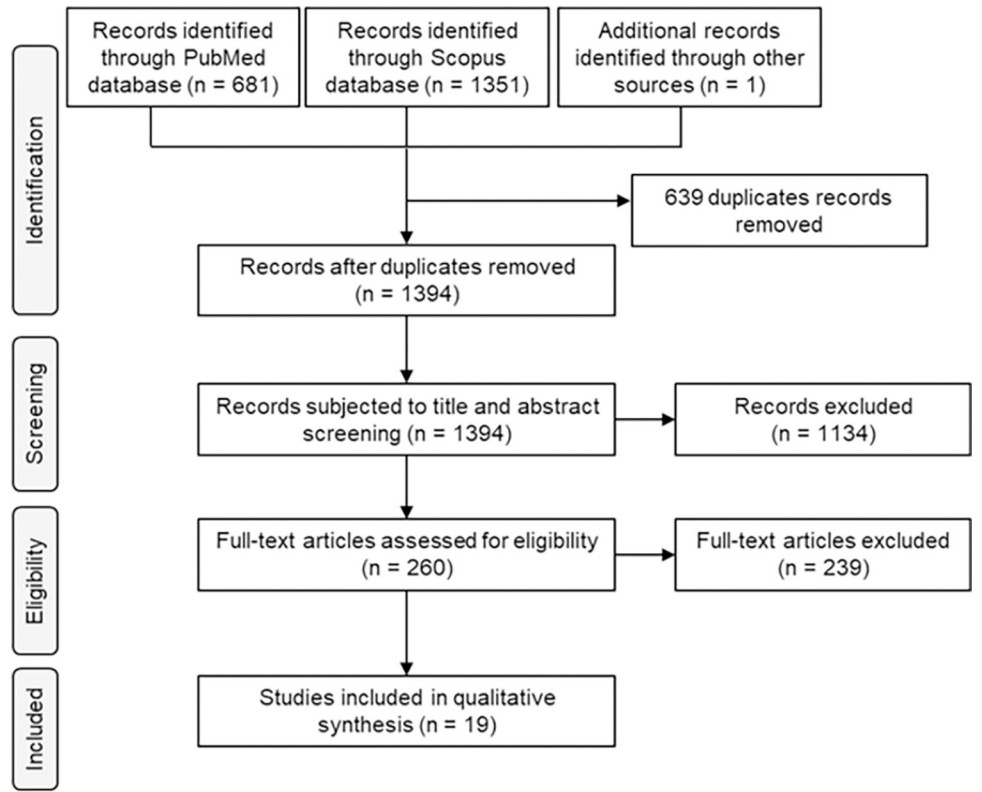

**Fig 1. Flow chart of record processing for inclusion in the systematic review.**

Among the 19 selected papers, lethal ovitraps were used in 15 studies, host-seeking female traps in 3 studies, and a combination of both in 1 study (Fig 2). In addition, the majority of these studies focused on the control of *Ae. aegypti* populations (16 of 19) and was conducted during periods with a low circulation of dengue and/or chikungunya and/or Zika viruses (10 out of 19) (Fig 2A). Regarding the design of the study, 5 were observational studies, 7 were randomized controlled trials and 7 were non-randomized controlled experimental trials (Fig 2B).

## Lethal ovitrap-based interventions

### Lethal ovitraps used to control *Aedes aegypti* populations

**Standard lethal ovitraps.** Standard Lethal Ovitraps (LOs) have been used to control *Ae. aegypti* populations in Brazil [21] and Thailand [22], in endemic areas for dengue. Both Perich *et al.* [21] and Sithiprasasna *et al.* [22] used the same LO consisting of a small 473-mL black plastic cup baited with a 10% hay (w/v) infusion and containing a 11 x 2.5 cm strip treated with deltamethrin. Neither study implemented other vector control actions in parallel (e.g., insecticide spraying, larviciding or mechanical control of larval breeding sites). LOs deployments resulted in a reduction of more than 40% in the abundance of females (collected by aspirator indoors) at one site at least, a reduction of 49% to 80% in larval positive containers and a reduction of 56% to 97% in the average number of pupae per house [21].

In addition, two similar studies, combining the use of LOs with other vector control methods, were conducted in eastern Thailand to reduce *Ae. aegypti* population [23,24]. Both studies were conducted in similar urban areas within a 100 m radius of dengue case foci located using geolocation tools. Kittayapong *et al.* [23] combined several actions to reduce *Ae. aegypti*

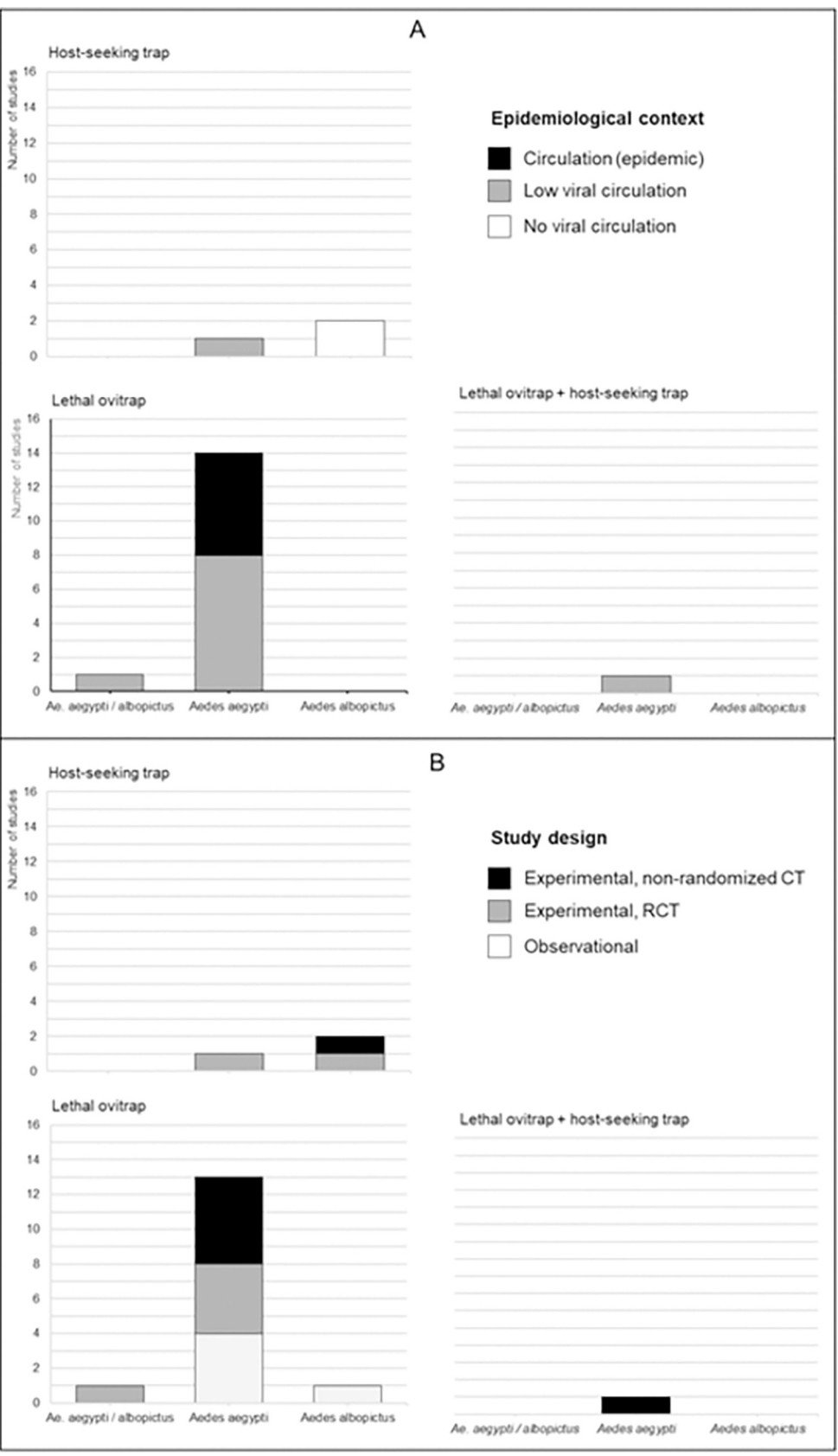

**Fig 2.** Distribution of selected articles according to the target species, the traps used and (A) the epidemiological context of the study (B) the study design. Lethal ovitrap includes AGO, GAT and Sticky traps.

populations: (i) removal of breeding sites before the rainy season, (ii) garbage collection during the rainy season, (iii) placement of mosquito net covers on domestic water jars, (iv) treatment of not removable breeding sites with locally produced *Bacillus thuringiensis israeliensis* (Bti), (v) seeding of larvivorous copepods (*Mesocyclops thermocyclopoides*) in permanent water containers, (vi) placement of LOs. The use of locally manufactured lethal ovitraps was the method used to reduce the adult vector population of *Ae. aegypti* that was not eliminated in the immature stages by all other vector control methods. The percentage of ovitraps that contained *Ae. aegypti* eggs decreased from 49.6% after the start of the intervention to 10.4% at the termination of the study. This impact can also be observed in confirmed cases of dengue hemorrhagic fever: 322 dengue cases per 100,000 inhabitants in the untreated area compared to zero cases in the treated area the year after the intervention. Subsequently, Kittayapong *et al*. [24] combined the same entomological interventions in the same study areas as previously [23]. These interventions were evaluated using anti-DENV IgM and IgG seroprevalence screening in six local schools with a population of approximately 1,800 students as serological criteria in addition to clinical criteria. The results of this study showed a significant reduction in: (i) *Ae. aegypti* density (35% containers positive for larvae before intervention vs. 1% after intervention and reduction of human landing collections to zero in the intervention area), (ii) the number of anti-DENV IgM-IgG positive children (13.5% pre-intervention vs. 0% post-intervention), and (iii) the number of clinical dengue cases (265 pre-intervention vs. zero post-intervention). The specific effect of LOs has not been not evaluated.

In Colombia, Ocampo *et al*. [25] studied the use of LOs combined or not with the use of slow-release briquette formulations of Bti larvicide as control methods of *Ae. aegypti*. The study was conducted in four similar neighborhoods (buffer zone around each site) in the city of Cali, with three treated neighborhoods (LOs, Bti and LOs + Bti) and one neighborhood without intervention. In all neighborhoods, including the non-intervention neighborhood, social mobilization and participation of the population in the elimination of breeding sites was carried out. The results showed a decrease in *Ae. aegypti* abundance after the interventions but in a similar way in the four neighborhoods studied indicating that an integrated strategy (with social mobilization and participation of the population in the elimination of breeding sites) for vector control offers better results than the exclusive use of one of the available tools (LOs, Bti or LOs + Bti).

In Australia (Cairns), a series of mass trapping interventions for *Ae. aegypti* were evaluated according to entomological criteria [26] and population acceptability [20]. The first two interventions, conducted during the dry and wet seasons, were a controlled trial with three conditions: (i) a combination of LOs, BG-Sentinel and reduction of larval breeding site (ii) reduction of larval breeding site reduction only, and (iii) no intervention area [26]. Mosquitoes collected by the BG-Sentinel set in intervention area were also used as an *Aedes* population measure with regular ovitraps (dry season) or sticky ovitraps (SOs) (wet season). In the untreated control area, 15 BGS traps and regular ovitraps (dry season) or SOs (wet season) were also used to assess the adult population. The third intervention was a controlled trial conducted during the wet season to evaluate the entomological efficacy of biodegradable lethal ovitraps (BLOs). Finally, a cross-sectional study was conducted during both the dry and wet seasons in order to assess the acceptability (measured as the failure of residents to remove LOs and BLOs set in their yards) and the performance of lethal ovitraps (LOs and BLOs) [26].

Interventions combining LOs and BG-Sentinel traps significantly reduced adult female populations during the wet season only, compared to the other conditions (non-intervention area and larval control only). Indeed, the weekly mean number of *Ae. aegypti* females per sticky ovitraps (used to monitor gravid females) decreased from 87% after the intervention. The use of BLOs significantly decreased the density of host-seeking females captured by

BG-Sentinel traps during the wet season, but the effect on the density of gravid females monitored by sticky ovitraps (SOs) was not significant [26]. The intervention and non-intervention areas were adjacent, so mosquito migration from one to the other could not be excluded. Regardless of the type of LOs, population acceptability was found to be positive (with <9% of traps missing after 4 weeks) during the intervention period [20], which was nevertheless limited to 4 weeks.

**Autocidal Gravid Ovitrap (CDC-AGO).**   Among the 19 articles selected in this scoping review, 9 articles concern studies conducted in Puerto Rico by the same authors between 2011 and 2016 to control *Ae. aegypti* and reduce arbovirus transmission on the island. These studies are related to the evaluation of the entomological and epidemiological efficacy of Autocidal Gravid Ovitrap (AGO) produced by the U.S. Centers for Disease Control and Prevention (CDC).

A first longitudinal study conducted since 2011, presents an evaluation of the efficacy of AGO traps in reducing *Ae. aegypti* populations in an urban area (La Margarita) compared to a reference site located 20 km away (Villodas) in which no traps were set [27]. In both sites, vector control interventions were conducted (reduction of larval breeding sites, population awareness, and larviciding). At La Margarita, 81% of houses were equipped with three AGO, from December 2011 to February 2012 and then with four AGOs, from March to October 2012 (total between 812 and 1050 AGOs). The results of this first entomological study revealed that mass trapping reduced *Ae. aegypti* females for 11 months of observation (between December 2011 and October 2012) in the area equipped with traps (La Margarita) compared to the reference area (Villodas). After allowing for rainfall and temperature effects, the relative reduction of female *Ae. aegypti* in the intervention area was plotted and indicated a reduction of 53% to 70% for females caught by BG-Sentinel and Stationary AGO (SAGO) traps, respectively.

A second publication reveals that trapping was continued in La Margarita until February 2014 and traps were also set in Villodas from February 2013 to February 2014 at a rate of 3 AGOs per house [28]. Therefore, catches made in 2011 and 2012 in Villodas could be considered as baseline data for this site. Two non-intervention sites (Playa and Arboleda) completed the setup of this second study. Three entomological indicators were explored in this study. The first indicator aimed to assess the reduction of *Ae. aegypti* density in Villodas after the deployment of mass trapping (2013–2014 comparison with 2011–2012 baseline data). The average number of *Ae. aegypti* females per SAGO trap per week was reduced by 79% compared to the baseline data collected at the same site in 2011–2012. The second indicator aimed to evaluate the difference in the reduction of *Ae. aegypti* density in Villodas and La Margarita sites in 2013–2014, considering that the mass trapping intervention started in 2011 in La Margarita while it started in 2013 in Villodas. The average catch in Villodas were lower than those measured in La Margarita during the same period. Finally, a last indicator showed that the average number of *Ae. aegypti* females caught per trap per week was 88% higher in the two non-intervention sites (Playa and Arboleda) compared to the two sites with traps (La Margarita and Villodas) [28].

Two other studies completed the results of the two previous entomological studies by investigating the density and infection rates of dengue and chikungunya viruses in *Ae. aegypti* in 2014 [29], followed by dengue, chikungunya, and Zika viruses in the same vector species in 2016 [30]. These studies were conducted after the consecutive emergence of chikungunya in May 2014 and Zika in December 2015 in Puerto Rico. As in the previous study, the two sites with intervention (e.g. 3 AGOs per house) were La Margarita and Villodas and the two control sites without intervention were Playa and Arboleda. The results of these studies showed that the average number of females caught/trap/week and the mosquito infection rates were much lower in the intervention sites with traps than those measured in the non-intervention sites. In

2014, results show a reduction of 91% in females caught and a 91% reduction in CHIKV infection rate at the sites with traps during the 7 months of observation (June-December 2014) [29]. However, no DENV-infected mosquitoes were collected at any of the 4 study sites during this period [29]. In 2016, the results showed (i) a 90% reduction in females captured at intervention sites; (ii) 2 pools of DENV-infected mosquitoes at the intervention sites compared to 4 at the non-intervention sites; (iii) 5 pools of CHIKV-infected mosquitoes at the intervention sites compared to 50 at the non-intervention sites; and (iv) 3 pools of ZIKV-infected mosquitoes in the intervention sites compared to 55 at the non-intervention sites, during the 12 months of observation (January-December 2016) [30].

In parallel, after the emergence of chikungunya in Puerto Rico in May 2014, a fifth epidemiological study was conducted from November 2015 to February 2016. This study aimed to assess the anti-CHIKV IgG seroprevalence among a sample of 28% of residents randomly drawn from communities in trap-equipped sites (La Margarita and Villodas) compared to communities in sites without intervention (Playa and Arborela) [31]. We note that the authors consider these seroprevalence data as incidences, given the emergence of CHIKV in a naive population in an insular context. The results showed that the proportion of anti-CHIKV IgG antibodies in communities at trap sites was half that in communities at non-intervention sites, during the 4-month observation period [31]. Seroprevalence results (anti-CHIKV IgG and IgM) obtained between November 2015 and January 2016 were the subject of a more comprehensive statistical analysis published in 2019 [32]. After calculating adjusted prevalence ratios, it was estimated that 26.1% of the 175 residents in sites equipped with AGO traps were infected with CHIKV while 43.8% of 152 residents in non-intervention sites were infected [32]. This difference in seroprevalence was observed when *Ae. aegypti* densities were ten times lower in the AGO trap intervention sites than in the non-intervention sites (see also [30]).

In addition, a complementary study on the feasibility and long-term acceptability of this mass trapping technique (but without entomological estimations of the efficacy) was conducted at La Margarita and Villodas sites between 2013 and 2015 [19]. This study showed that since the initial deployment of AGOs in December 2011 in La Margarita and in February 2013 in Villodas, most residents kept the traps in their properties and allowed them to be collected every 2 months [19]. The percentage of houses with 3 AGOs traps in La Margarita was 85–87% and 83–87% in Villodas.

Other studies have also been conducted in Puerto Rico in sites other than La Margarita and Villodas. A longitudinal study conducted in El Coco during the 2016 Zika emergence [33] showed that in an Integrated Vector Control (IVC) based on: (i) community awareness and education, (ii) source reduction of larval breeding sites and larviciding, and (iii) mass-trapping with 3 AGOs per house applied to at least 80% of the houses in an area, resulted in a 92.4% reduction of mosquitoes per trap per week in an area equipped with traps [33].

In addition, a cluster randomized stepped-wedge design trial was conducted in 2016 at 8 sites to evaluate the efficacy of IVC in limiting Zika virus transmission during the 2016 outbreak in the city of Caguas [34]. No significant reduction in mosquito numbers was observed when control coverage was between 0 and 20%, and the reduction became significant when control coverage increased from 21 to 40% (34.3% reduction), 41 to 60% (42.4% reduction), 61 to 80% (62% reduction), and >80% (81.5% reduction). However, the low incidence of Zika virus did not allow them to evaluate the impact of trap interventions on virus transmission during the study [34].

### *Sticky traps*-based interventions

In Manaus, Brazil, the efficacy of MosquiTRAP (MQT) in reducing *Ae. aegypti* populations was evaluated in a context of low dengue viral circulation [35]. A Cluster randomized

controlled trial (cRCT) was conducted in an urban residential area for 17 months (February 2009-June 2010). Clusters consisted of approximately 100 to 150 households with 3 sticky traps per household in the intervention areas placed outdoors in the peri-domestic area. The evaluation of these interventions was carried out according to entomological and epidemiological criteria (Table 3). The use of MosquiTRAP did not reduce the abundance of *Ae. aegypti* or the number of dengue cases in the area equipped with traps [35].

### Lethal ovitraps used to control *Aedes albopictus* populations

A citizen-based intervention was conducted in University Park, USA to reduce the nuisance of *Ae. albopictus* [36]. City residents were encouraged to purchase two BG-GATs per house and maintain them during the intervention. 439 neighborhoods in the city were equipped with BG-GATs, with different coverage by neighborhood. The results of this study showed that in neighborhoods where more than 80% of households were equipped with BG-GATs, the number of *Ae. albopictus* females captured by BG-Sentinel monitoring was significantly lower than that measured in neighborhoods with 80% or less coverage [36].

## Host-seeking female trap-based interventions

### Host-seeking female traps used to control *Aedes aegypti* populations

In Manaus, Brazil, a Cluster randomized controlled trial (cRCT) was conducted to evaluate the efficacy of BG-Sentinel traps (without $CO_2$) in reducing *Ae. aegypti* populations [37]. The intervention was carried out in a situation of low dengue viral circulation (inter-epidemic context). This study was conducted in an urban residential area for 17 months (February 2009-June 2010), including at least one wet season (period of high mosquito activity) and one dry season (period of low mosquito activity) (Table 4). The clusters were composed of approximately 100–150 households and the intervention achieved an overall coverage of approximately 60% of households. The BG-Sentinel traps intervention significantly reduced (54% reduction) the mean number of *Ae. aegypti* females caught/trap/24 h in the areas equipped with traps only during the first wet season. No significant effects were observed thereafter until the end of the entire 17-month observation period. The serological survey did not reveal any significant differences in dengue infections between the areas with traps and the control areas [37].

### Host-seeking female traps used to control *Aedes albopictus* populations

Englbrecht *et al.* [38] conducted an evaluation of the efficacy of BG-Sentinel traps in controlling local *Ae. albopictus* populations in Cesena in northern Italy. A non-randomized controlled trial was conducted and 3 intervention sites were paired with 3 comparable non-intervention sites. The intervention sites were: (i) a single-family house with a garden equipped with 7 BG-Sentinel traps placed outdoors, (ii) a cemetery equipped with 8 BG-Sentinels, and (iii) an apartment area equipped with 8 BG-Sentinels placed outdoors. The traps were spaced 5–10 m apart with a density ranging from 1 trap/150 m$^2$ to 1 trap/350 m$^2$. Five weeks after the intervention, the number of *Ae. albopictus* captured during the 1.5 hours of weekly human landing collections (HLC) at the intervention sites was significantly lower than that measured at the non-intervention sites. Over the entire study period, an average reduction of 87% of *Ae. albopictus* collected by HLC was obtained at the intervention sites equipped with BG-Sentinel traps [38].

In southern France, Akhoundi *et al.* [39] studied the effect of the "BioBelt anti- Mosquitoes" barrier trap on *Ae. albopictus* biting nuisance. The system, BioBelt Anti-Mosquitoes, is a

**Table 3. Summary of lethal ovitrap-based interventions to control *Aedes* populations.**

| Reference | Epidemio-ogical context | Target species | Geographic location | Experimental protocol | Study duration \| Period of the year | Used trap \| Attractant \| Killer agent | General information on study design | Number of traps \| Coverage (% of treated houses) | Indicators used to measure trapping efficacy | Effect measured |
|---|---|---|---|---|---|---|---|---|---|---|
| | | | | | | *Standard Lethal Ovitraps (LOs)* | | | | |
| Perich *et al.* (2003) [21] | Endemic circulation of DENV | *Ae. Aegypti + Ae. albopictus* | Brazil (Areia Branca and Nilopolis—State of Rio de Janeiro) | Cluster randomized controlled trial | 12 weeks \| February to May 2001 | LO \| Hay \| deltamethrin (1mg/strip) | Intervention area: group of 30 houses with LO Control: A group of 30 houses without LOs Both areas are located in the same neighborhood: no buffer zone between intervention and control houses | 10 LOs per house (5 traps inside and 5 outside each house) \| 100% | Percentage of containers positive for larvae and/or pupae Total pupae/house Number of *Aedes* females collected inside houses using aspirators | **Entomological effect.** Post-intervention density of *Ae. aegypti* was significantly reduced for most indicators (p<0.01), as shown by fewer positive containers (4–5 vs. 10–18) and pupae/house (0.3–0.7 vs. 8–10) at LO-treated vs. untreated houses, 3 month post-treatment at both municipalities. The mean number of *Ae. aegypti* females indoors were consistently reduced in LO-treated houses at Areia Branca (3.6 vs. 6.8/houses 3 months post-intervention) but not at Nilopolis (3/house, attributed to immigration). Very few *Ae. albopictus* were found and this species was excluded from the assessment. **Epidemiological effect.** No indicators |
| Sithiprasasna *et al.* (2003) [22] | Endemic circulation of DENV | *Ae. aegypti* | Thailand (Chom Bung District, Ratchaburi Province) | Cluster randomized controlled trial | 30 weeks \| Two studies: from 7 April to 29 October in 1999 and from 24 May to 15 December in 2000 | LO \| Hay \| deltamethrin | Intervention area: group of 50 houses with LO Control: group of 50 houses without LOs Both areas are located in the same village but separated by 250 m: buffer zone between intervention and control houses | 10 LOs per house (5 indoors and 5 outdoors) \| 100% | Mean number of containers positive for *Ae. aegypti* larvae and/or pupae Number of *Ae. aegypti* females collected inside houses using aspirators | **Entomological effect.** In 1999 (first study): the LOs had a negligible impact on all measures of *Ae. aegypti* abundance that were assessed. In 2000 (second study): a significant suppression of the local *Ae. aegypti* population was achieved as assessed by the reduction of multiple entomological indicators: 47% reduction in the abundance of *Ae. aegypti* female adult; 49% reduction in containers containing *Ae. aegypti* larvae; 56% reduction in containers containing *Ae. aegypti* pupae. **Epidemiological effect.** No indicators |
| Kittayapong *et al.* (2006) [23] | Endemic circulation of DENV | *Ae. aegypti* | Thailand (Plaeng Yao District, Chachoengsao Province) | Non-randomized controlled trial | 8 months \| - | LO \|—\| Permethrin | LO campaign was associated with an integrated vector control program. Intervention area: a group of houses located within 100 m of dengue cases Control: a group of houses located within a 100 m radius without dengue cases | 2–5 LOs per house (total of 406 traps) \| 100% | Percentage of positive ovitraps with *Ae. aegypti* eggs | **Entomological effect.** The specific action of LOs has not been evaluated. The percentage of ovitraps that contained *Ae. aegypti* eggs decreased from 49.6% after the start of the intervention to 10.4% at the termination of the study. **Epidemiological effect.** No indicators |
| Kittayapong *et al.* (2008) [24] | Endemic circulation of DENV | *Ae. aegypti* | Thailand (Plaeng Yao District, Chachoengsao Province) | Non-randomized controlled trial | 8 months \| - | LO \|—\| Permethrin | LO campaign was associated with an integrated vector control program. Intervention area: group of houses located within 100 m of dengue cases Control: group of houses located within a 100 m radius without dengue cases | 5–10 LOs per house \| 100% | Percentage of containers positive for *Ae. aegypti* larvae Number of *Ae. aegypti* females collected per house by weekly Human Landing Catch (HLC) Prevalence of anti-DENV IgG-IgM among students in control and intervention areas | **Entomological effect.** The specific action of LOs has not been evaluated. The number of *Ae. aegypti*-positive containers decreased from 35% to 1% in the intervention areas. The mean number of *Ae. aegypti* females was significantly reduced in the intervention areas compared to the control areas. **Epidemiological effect.** The proportion of anti-DENV IgG–IgM positive students in the intervention areas were reduced from 13.46% to 0% while those in control areas increased from 9.43% to 19.15%. |
| Ocampo *et al.* (2009) [25] | Endemic circulation of DENV | *Ae. aegypti* | Colombia (Cali) | Non-randomized controlled trial | 4 months \| April-July 2005 | LO \|—\| deltamethrin (0,1 mg/strip) | The study was implemented in four neighborhoods. Four treatments were evaluated in blocks within each neighborhood: (1) LO, (2) Bti, (3) LO and Bti and education and (4) one control (education only), as well as in a buffer area around each block Intervention area: 40 houses per block in an area of 100 m² Control: 40 houses per block in an area of 100 m² | 10 LOs per house (5 indoor and 5 outdoor) \| 100% | Percentage of houses positive for *Ae. aegypti* larvae and/or pupae Average number of *Ae. aegypti* pupae per house Number of houses infested with *Ae. aegypti* females indoors | **Entomological effect.** Results show no significant differences between interventions, and between intervention and control areas. However, the interventions (including education only as control) achieved a significant reduction in entomological indices compared with those observed during the pre-intervention survey: House index 15.1% vs. 8.5%, mean pupae per house 1.15 vs. 0.073, and Adult index 56.3% vs. 34.8% (p<0.05). **Epidemiological effect.** No indicators |

(*Continued*)

**Table 3.** (Continued)

| Reference | Epidemiological context | Target species | Geographic location | Experimental protocol | Study duration \| Period of the year | Used trap \| Attractant \| Killer agent | General information on study design | Number of traps \| Coverage (% of treated houses) | Indicators used to measure trapping efficacy | Effect measured |
|---|---|---|---|---|---|---|---|---|---|---|
| Rapley *et al.* (2009) [26] | Low circulation of DENV | *Ae. aegypti* | Australia (Cairns) | Non-randomized controlled trial | 8 weeks \| dry season (June to July 2006) | BG-Sentinel \| BG-Lure \| - + LO \|—\| bifenthrin (13.5 cm x 5 cm strip) | Two interventions were evaluated: (1) LO and BG-Sentinel and larval control, (2) larval control only. Control was without any intervention. Treatments were implemented during a 4-week period, after a pre-intervention period of 4 weeks. Intervention area based on LOs and BG-Sentinel: a group of 72 houses located within 200 m of a hypothetical dengue case Intervention based on larval control in houses (>70) located within 200 m of a hypothetical dengue case Control: a group of houses located in the non-intervention area | 4 LOs per house (total 206) + 1 BG-Sentinel trap per house (total 15–16); placed outdoors \| LOs: 75%; BG-Sentinel trap: 21% | Abundance of *Ae. aegypti* eggs through ovitrap collections Number of *Ae. aegypti* females captured during 2 consecutive days each week (15 BG-Sentinel traps) | **Entomological effect.** No significant decrease in the number of *Ae. aegypti* eggs and females could be attributed to the intervention. **Epidemiological effect.** No indicators |
| | | | | | 8 weeks \| wet season (March to April 2007) | BG-Sentinel \| BG-Lure \| - + LO \|—\| bifenthrin (13.5 cm x 5 cm strip) | Interventions as above. Intervention area based on Los abd BG-Sentinel: group of 90 houses located within 200 m of dengue cases Intervention based on larval control alone in houses (>70) located within 200 m of a hypothetical dengue case Control: a group of houses located in the non-intervention area | 4 LOs per house (total 243) + 1 BG-Sentinel trap per house (total 15–16); placed outdoors \| LOs: 71%; BG-Sentinel trap: 18% | Number of *Ae. aegypti* females captured during 2 consecutive days each week (15 BG-Sentinel traps) Number of *Ae. aegypti* females captured per week using 20 sticky ovitraps (SOs) | **Entomological effect.** The mean number of *Ae. aegypti* females collected after 4 weeks with Sticky Ovitraps and BG-Sentinels traps was significantly reduced in the intervention areas compared to the control areas. **Epidemiological effect.** No indicators |
| | | | | | 8 weeks \| wet season, (19 February– 20 March 2008) | Biodegradable lethal ovitrap (BLO) \|—\| - | Intervention area: three treatment areas were each provided with BLOs (>500, ~4/premise) plus larval control (total 179 houses) Control: group of houses located in the non-intervention area | 3.3 (mean) BLOs per house (total 552) \| 93% | Number of *Ae. aegypti* females captured during 2 consecutive days each week (15 BG-Sentinel traps) Number of *Ae. aegypti* females captured per week using 20 sticky ovitraps (SOs) | **Entomological effect.** The mean number of *Ae. aegypti* females collected in the intervention areas were significantly reduced compared to the control areas for BG-Sentinels traps but not for Sticky Ovitraps. **Epidemiological effect.** No indicators |
| | | | | | *Autocidal Gravid Ovitrap (CDC-AGO)* | | | | | |
| Barrera *et al.* (2014) [27] | Endemic circulation of DENV | *Ae. aegypti* | Puerto Rico (La Margarita and Villodas) | Non randomized controlled trial | 48 weeks \| Baseline: oct-dec 2011; Follow-up of *Ae. aegypti* density: dec 2011-oct 2012 | CDC-AGO \| Hay \| Sticky glue board | Intervention area: La Margarita (18 ha, 327 houses) Control: Villodas (11ha, 241 houses) | 3 CDC-AGOs per house \| 81% | Number of *Ae. aegypti* females captured per week using BG-Sentinel or Stationary AGO (SAGO) traps | **Entomological effect.** The mean number of *Ae. aegypti* females was significantly reduced in the intervention area compared to the control area with 53% to 70% for females captured by BG-Sentinel and SAGO traps, respectively. **Epidemiological effect.** No indicators |
| Barrera *et al.* (2014) [28] | Endemic circulation of DENV | *Ae. aegypti* | Puerto Rico (La Margarita, Villodas, Playa and Arboleda) | Non randomized controlled trial | Margarita: 117 weeks \| dec 2011 —feb 2014 Villodas: 56 weeks \| feb 2013– 2014 | CDC-AGO \| Hay \| Sticky glue board | Intervention area: La Margarita (18 ha, 327 houses) and Villodas (11 ha, 241 houses) Control: Playa (17 ha, 269 houses) and Arboleda (21 ha, 398 houses) | 3 CDC-AGOs per house \| 85% | Number of *Ae. aegypti* females captured per week using Stationary AGO (SAGO) traps | **Entomological effect.** The mean number of *Ae. aegypti* females was significantly reduced by 79% after placing the AGO intervention traps in Villodas. The mean number of *Ae. aegypti* females after placing the AGO intervention traps in Villodas was lower in Villodas than in La Margarita during the same period. Areas with AGO intervention traps (Villodas and La Margarita) had 88% fewer *Ae. aegypti* females than nearby control areas (Playa and Arboleda). **Epidemiological effect.** No indicators |
| Barrera *et al.* (2017) [29] | Co-Circulation of DENV and CHIKV | *Ae. aegypti* | Puerto Rico (La Margarita, Villodas, Playa and Arboleda) | Observational (with untreated areas vs treated areas) | 7 months \| jun-dec 2014 | CDC-AGO \| Hay \| Sticky glue board | Intervention area: La Margarita (18 ha, 327 houses) and Villodas (11 ha, 241 houses) Control: Playa (17 ha, 269 houses) and Arboleda (21 ha, 398 houses) | 3 CDC-AGOs per house \| 85% | Number of *Ae. aegypti* females captured per week using Stationary AGO (SAGO) traps CHIKV infection rates of gravid *Ae. aegypti* females captured weekly in SAGO traps | **Entomological effect.** The mean number of *Ae. aegypti* females was 10.5 times lower (91%) in the two areas with AGO intervention traps compared to control areas during the study. CHIKV infection rates of gravid *Ae. aegypti* females were ten times higher (90.9%) in the control areas (50/55 CHIKV positive pools) than in the intervention areas (5/55). **Epidemiological effect.** No indicators |

*(Continued)*

**Table 3.** (Continued)

| Reference | Epidemio-ogical context | Target species | Geographic location | Experimental protocol | Study duration \| Period of the year | Used trap \| Attractant \| Killer agent | General information on study design | Number of traps \| Coverage (% of treated houses) | Indicators used to measure trapping efficacy | Effect measured |
|---|---|---|---|---|---|---|---|---|---|---|
| Barrera et al. (2019) [30] | Co-Circulation of DENV, CHIKV and ZIKV | Ae. aegypti | Puerto Rico (La Margarita, Villodas, Playa, Arboleda) | Observational (with untreated areas vs treated areas) | 2 years \| Every week during January-December in 2014 and 2016 | CDC-AGO \| Hay \| Sticky glue board | Intervention area: La Margarita (18 ha, 327 houses) and Villodas (11 ha, 241 houses) Control: Playa (17 ha, 269 houses) and Arboleda (21 ha, 398 houses) | 3 CDC-AGO per house \| over 80% | Number of Ae. aegypti females captured per week using SAGO traps CHIKV, ZIKV and DENV infection rates of gravid Ae. aegypti females captured weekly in SAGO traps | **Entomological effect.** The mean number of Ae. aegypti females were 5.5–9.5 times lower in the two areas with AGO intervention traps compared to control areas during the study. Infection rates of gravid Ae. aegypti females were much lower in the intervention areas than in the control areas: 50% reduction for DENV, 90% reduction for CHIKV, and 95% reduction for ZIKV. **Epidemiological effect.** No indicators |
| Lorenzi et al. (2016) [31] | Circulation of CHIKV | Ae. aegypti | Puerto Rico (Salinas and Guayama) | Observational (retrospective study) | 2012–2016 \| Seroprevalence surveys: nov 2015-feb 2016 | CDC-AGO \| Hay \| Sticky glue board | Intervention and control areas: 620 houses (28.5% of residents) Intervention area: 175 participants Control area: 152 participants | 3 CDC-AGOs per house \| 85% | Prevalence of anti-CHIKV IgG among residents (28% randomly sampled) of control and intervention areas | **Entomological effect.** No indicators **Epidemiological effect.** The proportion of anti-CHIKV IgM positive residents was reduced by one half in the intervention areas compared to control areas during the study. |
| Sharp et al. (2019) [32] | Circulation of CHIKV | Ae. aegypti | Puerto Rico (La Margarita, Villodas, Playa, Arboleda) | Observational (retrospective study) | Evaluation of AGO traps has been ongoing since 2012 Surveys were conducted between November 16, 2015 and January 16, 2016 | CDC-AGO \| Hay \| Sticky glue board | Intervention area: La Margarita (18 ha, 327 houses) and Villodas (11 ha, 241 houses) Control: Playa (17 ha, 269 houses) and Arboleda (21 ha, 398 houses) | 3 CDC-AGOs per house \| La Margarita: 88%; Villodas: 84% | Prevalence of anti-CHIKV IgG-IgM among residents of control and intervention areas | **Entomological effect.** No indicators **Epidemiological effect.** The proportion of anti-CHIKV IgG-IgM positive residents was estimated as 26.1% in the intervention areas compared to 43.8% in the control areas. |
| Barrera et al. (2018) [33] | Circulation of ZIKV | Ae. aegypti | Puerto Rico (El Coco) | Cluster randomized controlled trial | 7 months \| jun-dec 2016 | CDC-AGO \| Hay \| Sticky glue board | Intervention area: Arcadio (150 m radius, 179 houses) Control: Santa Ana (150m radius, 164 houses) | 3 CDC-AGOs per house \| 84% | Number of Ae. aegypti females captured per week using Stationary AGO (SAGO) traps ZIKV infection rates of gravid Ae. aegypti females captured weekly in Stationary AGO (SAGO) traps | **Entomological effect.** The mean number of Ae. aegypti females was significantly reduced in the intervention areas from 27.7 mosquitoes/trap/week before the intervention to 2.1 after (92.4% reduction). After covering areas initially without intervention with AGO traps (crossover trial), the mean number of Ae. aegypti females was significantly reduced from 22.4 to 3.5 (84.3% reduction). The low circulation of Zika virus in the study areas made it impossible to assess the impact of the intervention on the infection rates of gravid Ae. aegypti females. **Epidemiological effect.** No indicators |
| Barrera et al. (2019) [34] | Co-Circulation of DENV, CHIKV, ZIKV | Ae. aegypti | Puerto Rico (Caguas) | Cluster randomized controlled trial (stepped-wedge design, all the areas are treated at different moments) | 11 months \| Every week during October 2016 to August 2017 | CDC-AGO \| Hay \| Sticky glue board | CDC-AGO campaign was associated with an integrated vector control program. Intervention area: 23.1 Km2 (61,511 inhabitants and 25,363 houses) No control area | 3 CDC-AGOs or more are placed per home and in public areas separated by 50 m \| over 80% in most clusters | Number of Ae. aegypti females captured per week using Stationary AGO (SAGO) traps CHIKV, ZIKV and DENV infection rates of gravid Ae. aegypti females captured weekly in Stationary AGO (SAGO) traps | **Entomological effect.** The mean number of Ae. aegypti females was significantly reduced in the intervention areas by 82.3%. The reduction in the mean number of Ae. aegypti females is not significant when the coverage of houses by AGO traps in an area is below 20%. The reduction in the mean number of Ae. aegypti females becomes significant when the coverage rate of houses by AGO traps in an area increases with 34.3% reduction for 21–40% of houses covered, 42.4% reduction for 41–60%, 62% reduction for 61–80% and 81.5% reduction when the house coverage rate exceeds 80%. In areas without vector control, rates in gravid Ae. aegypti females of ZIKV detection in 2016 were significantly higher, similarly to those observed for CHIKV in 2014. No significant differences in infection rates with ZIKV and CHIKV at the same sites between the years were observed. **Epidemiological effect.** No indicators |

*Sticky traps*

(Continued)

**Table 3.** (Continued)

| Reference | Epidemio- logical context | Target species | Geographic location | Experimental protocol | Study duration \| Period of the year | Used trap \| Attractant \| Killer agent | General information on study design | Number of traps \| Coverage (% of treated houses) | Indicators used to measure trapping efficacy | Effect measured |
|---|---|---|---|---|---|---|---|---|---|---|
| Degener *et al.* (2015) [35] | Endemic circulation of DENV | *Ae. aegypti* | Brazil (Manaus) | Cluster randomized controlled trial | 17 months (15 months + two months of baseline monitoring \| 2 rainy seasons + 1 dry season | MosquiTRAP \| — \| Sticky | Intervention area: 3 intervention clusters Control: 3 non-intervention clusters 104 to 150 houses per cluster; with a mean of 129 houses and a total of 775 houses | 3 MosquiTRAP per household (placed outdoors in the peri-domestic area) \| 51.1% to 53% | Number of *Ae. aegypti* females captured using 4 BG-Sentinel traps every 15 days Prevalence of anti-DENV IgM among residents of control and intervention areas | **Entomological effect.** The intervention had no significant effect on the mean number of *Ae. aegypti* females. **Epidemiological effect.** The proportion of anti-IgG positive residents was equivalent in the intervention and in the control areas. |
| *Gravid Aedes Trap (GAT)* | | | | | | | | | | |
| Johnson *et al.* (2018) [36] | Absence of viral circulation | *Ae. albopictus* | United States (University Park, Maryland) | Observational (without control areas) | 2017 with preliminary trials in 2016 \| Late June to October 2017 | BG—GAT \|—\| Canola Oil | Intervention area: About 1,300 ha (approximately 1,000 residential yards) No control area but (comparison between blocks with low (<50%) and high (>50%) GATs coverage) | 2 BG-GATs per household (one in the backyard and one in the frontyard) \| Gradient of coverage from 0 to more than 80% | Number of *Ae. albopictus* females captured using BG-Sentinel traps (8 sampling events (24 hours each) over a six-week period separated in time by an average of 6.12 days) | **Entomological effect.** The mean number of *Ae. albopictus* females was significantly reduced in the intervention areas when the coverage rate of houses by BG-GATs traps is more than 80%. **Epidemiological effect.** No indicators |

commercial trap system developed and patented by the French company HBM Distribution SAS. It consists of a network of traps positioned at an average distance of 5 m around the area to be protected and connected to a control center [39]. Three houses with gardens, located in a residential area of Bar-sur-Loup (Provence-Alpes-Côte d'Azur) were treated with this system. Already in the first week, a 50% reduction of the biting nuisance was observed in the houses equipped with traps compared to non-intervention houses. After three weeks of continuous trapping, the biting nuisance was significantly reduced in the trap-protected houses compared to the non-intervention houses, and after six weeks, the biting nuisance was reduced to almost zero until the end of the 3-month trial [39].

## Discussion

Most of the articles analyzed in this review concern the use of LOs to control *Ae. aegypti* populations, in areas endemic for dengue and/or chikungunya and/or Zika viruses (Brazil, Colombia, Puerto Rico, Thailand). There is evidence of the efficacy of lethal ovitrap-based interventions that result in a reduction in mosquito vector populations (measured by adult females, larvae, or pupae) [21,22]. Moreover, the efficacy of LOs can increase under the following circumstances: i) when deployed at a density of more than three traps per house in an area where at least 60–80% of houses are equipped (coverage) [34] and ii) when used in combination with other vector control actions, particularly larval control interventions (i.e. suppression of breeding sites and larviciding) [23,24]. Of all the studies, only AGO trap trials in Puerto Rico provided evidence of reduced viral transmission (through reduced viral infection rates in mosquito females and serological incidences in humans). According to the studies with serologic indicators, infection rates of *Ae. aegypti* mosquitoes with chikungunya and Zika viruses were lower at sites equipped with traps than at control sites [29,30]. Similarly, a decrease in CHIKV transmission in areas where more than 80% of households were equipped with three AGOs traps was demonstrated by Lorenzi *et al.* [31] and Sharp *et al.* [32]. However, the strength of evidence of these results remains moderate because they may be subject to bias [40].

The studies of Englbrecht *et al.* [38] and Akhoundi *et al.* [39] have the advantage of using human landing collections (HLC) as an indicator, which allows a direct assessment of mosquito biting nuisance. Thus, these two studies provide strong evidence that the continuous use of BG-Sentinel traps in urban areas (at least 1 trap/house) or the use of "Biobelt anti-mosquitoes" trap barriers can significantly reduce in the trap-protected houses the average number of

**Table 4. Summary of host-seeking female trap-based interventions to control *Aedes* populations.**

| Reference | Epidemiological context | Target species | Geographic location | Experimental protocol | Study duration \| Period of the year | Used trap \| Attractant \| Killer agent | General information on study design | Number of traps \| Coverage (% of treated houses) | Indicators used to measure trapping efficacy | Effects measured and conclusions |
|---|---|---|---|---|---|---|---|---|---|---|
| Degener et al. (2014) [37] | Endemic circulation of DENV | Ae. aegypti | Brazil (Manaus) | Cluster randomized controlled trial | 73 weeks \| Rainy season 2008–2009, Dry season 2009, Rainy season 2009–2010 | BG-Sentinel \| — \| - | 6 intervention and 6 non-intervention clusters (103–151 households per cluster) | 1 BG-Sentinel per (about 26 traps / ha \| 60.5% | Number of Ae. aegypti females captured using 4 BG-Sentinel traps every 15 days Proportion of Ae. aegypti parous females caught in the BG-Sentinel traps (age estimation of the target population) | **Entomological effect.** Entomological monitoring indicated that BG-Sentinel mass trapping significantly reduced the abundance of Ae. aegypti females during the first five rainy months (reduction of 54% before/after). During the next dry season, when the mosquito population was lower, no effect of BG-Sentinel mass trapping was observed. During the first rainy season, the proportion of parous females was significantly lower in the intervention areas than in the control areas. Fewer Ae. aegypti females were measured in the intervention clusters during the next rainy period, but no significant difference between clusters were observed. **Epidemiological effect.** No indicators |
| | Endemic circulation of DENV | Ae. aegypti | Brazil (Manaus) | Cluster randomized controlled trial | 73 weeks \| Rainy season 2010 | BG-Sentinel \| — \| - | 6 intervention and 6 non-intervention clusters (103–151 households per cluster) | 1 BG-Sentinel per house (about 26 traps / ha \| 60.5% | Prevalence of anti-DENV IgM among residents of control and intervention areas | **Entomological effect.** No indicators **Epidemiological effect.** In houses participating in BG-Sentinel mass trapping in the intervention areas, recent dengue infections (IgM) among residents were less frequent than in the control areas, although this effect was not statistically significant. |

(*Continued*)

**Table 4.** (Continued)

| Reference | Epidemiological context | Target species | Geographic location | Experimental protocol | Study duration \| Period of the year | Used trap \| Attractant \| Killer agent | General information on study design | Number of traps \| Coverage (% of treated houses) | Indicators used to measure trapping efficacy | Effects measured and conclusions |
|---|---|---|---|---|---|---|---|---|---|---|
| Englbrecht et al. (2015) [38] | Absence of viral circulation | Ae. albopictus | Italy (Cesena, Emilia–Romagna region) | Non-randomized controlled trial | 16 weeks \| from the end of June to October 2008 | BG-Sentinel \| BG-Lure \| - | 3 intervention sites and 3 non-intervention sites | 7 or 8 traps /site | Number of Ae. albopictus females collected by weekly Human Landing Catch (HLC) Abundance of Ae. albopictus eggs through ovitrap collections | **Entomological effect.** The mean number of Ae. albopictus females was significantly reduced in the intervention areas compared to the control areas, with a reduction of about 87% in the Human Landing Catch (HLC). The number of Ae. albopictus eggs collected in ovitraps was significantly reduced in the intervention areas compared to the control areas (reduction of 64%). **Epidemiological effect.** No indicators |
| Akhoundi et al. (2018) [39] | Absence of viral circulation | Ae. albopictus | France (Le Bar-sur-Loup; Provence-Alpes-Côte d'Azur) | Cluster randomized controlled trial | 3 months \| July to September 2016 | BioBelt Anti-Mosquitoes \| $CO_2$+ BG-Lure \| - | 3 intervention houses and 3 non-intervention houses | 9, 13 and 18 traps installed to protect the three intervention houses \| - | Number of Ae. albopictus females collected by weekly Human Landing Catch (HLC) performed during 30 min | **Entomological effect.** The trap barrier system was very effective in reducing the Ae. albopictus biting rates to almost zero in the trap-protected houses 6 weeks after the beginning of the intervention. The reduction in biting rates in the trap-protected houses reached 50% one week after the start of the intervention, 75% after 5 weeks and up to almost 100% 6 weeks post intervention until the end of the 3-month trial. **Epidemiological effect.** No indicators |

female *Ae. aegypti* [38] as well as the average number of *Ae. albopictus* [38,39] within a few weeks of their installation. These studies were carried out in areas with low viral circulation or in the absence of *Aedes*-borne diseases, which does not allow them to conclude that these traps are effective in stopping or even reducing arbovirus transmission.

The effect of all of these trap-based interventions on the reduction in mosquito vector populations is observed within a few weeks of deployment, but it is not possible to know how long it takes to observe an effect on virus transmission. It is important to note that in most cases, evidence of the efficacy of a mass trapping intervention following a rigorous testing protocol is scarce to date. Largely speaking, studies evaluating the efficacy of vector control interventions lack rigor in key aspects such as design, operationalization, data management and data analysis [41]. Guidance has been proposed to improve the overall conduct of field trials of vector control tools and strategies [40]. This increased rigor and standardization, should be similar to how large-scale Phase III clinical trials are conducted and should be of a multidisciplinary nature by including entomological as well as serological, virological and epidemiological outcomes to assess the efficacy of the intervention.

Few comparative studies exist regarding their design and context (such as cluster randomized controlled trials (CRCTs) targeting *Ae. aegypti* in endemic areas of viral circulation): two studies for LOs [21,22], two for AGOs [33,34], one for sticky traps [35] and one for host-seeking BG-Sentinel traps [37]. Furthermore, these studies assessed trapping efficacy using different indicators (number of host-seeking females, number of gravid females, proportion of positive containers, virus infection rate in mosquito females or serological studies among residents). Therefore, these limits do not allow for the development of an analysis or meta-analysis of efficacy among particular CRCTs studies.

We consider that based on the available data, mass trapping can be considered a promising complementary vector control tool that can be effective in combination with other control methods (including community mobilization or Sterile Insect Techniques e.g., SIT) and with good implementation quality (high house coverage, a sufficient number of traps per house, acceptability and sustainability). The use of traps in a preventive context allows for the reduction of mosquito vector populations density in urban areas located in epidemic or endemic regions, especially where adulticide treatment is not feasible (especially where there are exclusion zones for insecticide treatment: proximity to a river, lake shores, hospitals, insectarium, etc.). In addition, in case of operational failure of adulticide treatments due to the development of insecticide resistance in target mosquito populations [42,43] mass trapping is an alternative to other non-insecticide control tools, such as Sterile Insect Techniques (SIT). However, in the mass trapping strategy, the deployment and maintenance of a large number of traps represent significant logistical and financial investments. Host-seeking female traps have the advantage of a relatively high capture rate compared to lethal ovitraps LOs [44,45]. However, these traps can be more expensive, as they require electricity and often a $CO_2$ source, which imposes constraints related to the transport, protection and safety issues of gas cylinders when used as a $CO_2$ source. Thus, the logistical constraints of host-seeking female traps might limit their use in mass trapping strategies in some circumstances. On the other hand, despite a relatively low capture rate compared to host-seeking female traps, LOs have a lower unit price and may be easier to deploy in large numbers. Indeed, these devices do not require electricity or $CO_2$ to operate. Large-scale intervention strategies use both LOs and host-seeking female traps because of their good acceptability [19,20]. In an integrated vector control context, social mobilization programs are known to strongly support the strategic approach. Moreover, public awareness campaigns focusing on the importance of trap maintenance can lead to further reductions in *Aedes* populations through, for example, the removal of breeding sites by residents [36].

It should be noted that the efficacy of LOs may depend on the number of breeding sites in the intervention area. The fewer breeding sites there are, the more attractive this type of trap will be. Therefore, any control strategy based on the use of LOs should be combined with population mobilization to source reduction of breeding sites. In addition, LOs can provide additional breeding sites if left unattended. Indeed, maintenance and monitoring of the trap network is essential to its efficacy. Furthermore, the impact of host-seeking female traps and LOs on non-target species is poorly documented, with only a few studies [46]. Therefore, further studies are needed to assess the potential effect of these traps on non-target fauna (arthropods, amphibians, reptiles, etc.).

The evaluation of the efficacy of *Aedes* control methods such as mass trapping is not straightforward but can be done. Although WHO has produced a guideline entitled '*Efficacy-testing of traps for control of Aedes spp. mosquito vectors*' [47], our review showed that there is a wide variety of methods used to assess the efficacy of traps. Actually, there is no standardized method to evaluate the entomological and epidemiological efficacy of these innovative control methods and few trial results are currently published or available. Our descriptive analyses highlight the importance of homogenizing the methodology, such as indicators, study design and calculation of the efficacy of the different trapping interventions. Indeed, the use of a common indicator and study design could facilitate the standardized comparison of different trapping interventions and provide a decision-support indicator for a successful vector control strategy. As larval and eggs density are not always directly related to adult density [47–50], WHO recommends that the assessment of the efficacy of traps as a method of *Aedes* control should be carried out with adult density as an indicator. Larval and egg density can be considered, but as a secondary indicator [47]. In addition, monitoring of epidemiological indicators in endemic or epidemic areas should be prioritized as the effectiveness of vector control must ultimately be assessed by the reduction in the disease incidence [51]. This review highlights a knowledge gap in demonstrating the efficacy of mass trapping of vector mosquitoes in reducing viral transmission and disease. Thus, further large-scale Phase III Cluster randomized controlled trials conducted in endemic areas and including epidemiological outcomes [52] are needed to establish scientific evidence for the reduction of viral transmission risk by mass trapping targeting gravid and/or host-seeking female mosquitoes.

Our work opens with several perspectives. Firstly, the protocols for evaluating the effect of traps should be standardized by following the recommendations of the World Health Organization [47], and following the most rigorous standards of clinical trials [40] in order to be able to compare the efficacy found in different studies. Secondly, the efficacy of the traps under operational deployment conditions should be studied in order to determine the optimal conditions for the deployment (house coverage, number of traps per house) under different epidemiological scenarios to achieve a significant reduction in mosquito's populations and viral transmission. Moreover, it is important to encourage an integrated approach including source reduction or other innovative alternative control methods [53] when evaluating the effectiveness of traps, taking into account the sustainability and social acceptability of these tools. Therefore, modeling approaches can be used to optimize the combined use of different control tools as part of a large-scale, operationally conducted integrated vector control strategy. These tools can be very useful to specify the most effective combinations and the conditions for their use in terms of temporality and spatial scale, depending on the climatic, environmental and epidemiological contexts [54].

All these considerations should be systematically addressed in existing international websites on vector control trials as the WHO Vector Control Advisory Group (VCAG). A set of information such as the type of intervention, target geographic area, objectives, endpoints, estimated duration, sample size and contact information is essential. This approach is used by

most credible research teams working on human clinical trials and is becoming a regulatory requirement in many countries. These resources help scientific groups identify gaps that need to be filled, avoid duplication of efforts and provide a solid foundation for creating synergies. The rigor of these trials can be assessed by the quality of these protocols. While we consider that commercial strategies for positioning trapping systems in a given market should be addressed at the national level, such labeling would encourage manufacturers to carry out standardized and rigorous evaluations to avoid the rollout of commercial trapping systems that have not been properly tested for efficacy. Finally, we emphasize that disease control programs must be evaluated and monitored. Evaluation of these programs in terms of public health is straightforward but should be carried out in conjunction with an economic evaluation to measure the cost-effectiveness and cost-benefit of different methods (traps, larviciding, adulticiding, etc.) and vector control strategies. An integral and constant evaluation allows for a better understanding and grants the adjustment of measures and their adaptation to local conditions to expand the impact of these programs. Together, these data can provide a body of evidence-based information that can guide cost-effective and sustainable vector control strategies, thereby reducing the disease burden and economic impact of vector-borne diseases. The pertinent scrutiny of these elements allows decision-makers to consider the appropriate approach and ensures the success of an integral strategy.

## Author Contributions

**Methodology:** Ali Jaffal, Johanna Fite, Thierry Baldet, Pascal Delaunay, Frédéric Jourdain, Ronald Mora-Castillo, Marie-Marie Olive, David Roiz.

**Validation:** Ali Jaffal, Johanna Fite, Thierry Baldet, Pascal Delaunay, Frédéric Jourdain, Ronald Mora-Castillo, Marie-Marie Olive, David Roiz.

**Writing – original draft:** Ali Jaffal.

**Writing – review & editing:** Johanna Fite, Thierry Baldet, Pascal Delaunay, Frédéric Jourdain, Ronald Mora-Castillo, Marie-Marie Olive, David Roiz.

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
