## [Decision Letter · Decision Letter 0]

9 May 2022

Dear Ms FITE,

Thank you very much for submitting your manuscript "Current evidence on the effectiveness of mosquito mass-trapping interventions to control Aedes populations and to reduce the transmission of Aedes-borne diseases: A systematic review" for consideration at PLOS Neglected Tropical Diseases. As with all papers reviewed by the journal, your manuscript was reviewed by members of the editorial board and by several independent reviewers. In light of the reviews (below this email), we would like to invite the resubmission of a significantly-revised version that takes into account the reviewers' comments. 

We cannot make any decision about publication until we have seen the revised manuscript and your response to the reviewers' comments. Your revised manuscript is also likely to be sent to reviewers for further evaluation.

Sincerely,

Olaf Horstick, FFPH(UK)

Associate Editor

Karin Kirchgatter

Deputy Editor

Reviewer's Responses to Questions

**Key Review Criteria Required for Acceptance?**

**Methods**

-Are the objectives of the study clearly articulated with a clear testable hypothesis stated?

-Is the study design appropriate to address the stated objectives?

-Is the population clearly described and appropriate for the hypothesis being tested?

-Is the sample size sufficient to ensure adequate power to address the hypothesis being tested?

-Were correct statistical analysis used to support conclusions?

-Are there concerns about ethical or regulatory requirements being met?

Reviewer #1: According to authors only 2 databases were used from their searches namely Pubmed and Scopus. The authors should explain in more details why they did not utilize other databases for their searches. 

Eligibility criteria. This is not clear to me. No reference is made on any exclusion criteria used for this systematic review while criteria 4 and 6 are both on the study type (maybe consider merging?). 

Lines 155-156. The authors should further expand on whether any tool was used for the quality assessment of the included studies.

Reviewer #2: The authors present a systematic review following PRISMA guidelines of published evidence on the effectiveness of mosquito mass-trapping interventions to control Aedes populations and to reduce the transmission of Aedes-borne diseases.

Reviewer #3: - This paper is more a scoping review than a systematic review

- Eligibility criterium: 

o Field studies: In the title, effectiveness is mentioned, but in some of the papers it looks more as an efficacy study – evaluating effect in 3 houses is not an effectiveness study. Hence, a better definition of ‘field studies’ is needed – suggestion to call it ‘community studies’

o Observational studies: some of the observational studies included were not ‘analytical study that quantify the effect of an outcome’, such as reference 22 and 29. Coverage is a process indicator and not an outcome of effectiveness indicator

- Data extraction and analysis: very good to have seasonality and coverage included as variables extracted

**Results**

-Does the analysis presented match the analysis plan?

-Are the results clearly and completely presented?

-Are the figures (Tables, Images) of sufficient quality for clarity?

Reviewer #1: Table 2. Authors should present in a clearer way how the 21 included studies are cited in the summary tables 2 and 3 Please consider adding the name of authors instead of reference and restructuring the table for a clearer reference of each included study. Why there is no reference of the main findings of each included study, apart from the reduction achieved? This is important in order to more clearly present the policy perspectives of this study.

Reviewer #3: - In the introduction, there is a clear presentation of 2 categories of traps – but this clear difference is disappearing in the methods/results section (especially the latter). Suggestion to improve structure of paper, following these two categories: give summarized result per categorie and afterwards give the stratified result per sub-categorie. Now it reads as 5 different result-stories, where the reader looses the oversight.

- A restructure into the following could improve clarity:

o Resuming effectiveness estimates on entomological indicators (differentiate if used as a sole measures or as part of a combined strategy)

o Resuming effectiveness estimates on epidemiological parameters (differentiate if used as a sole measures or as part of a combined strategy)

o Resuming factors influencing the effectiveness of the tools

- Another general comment on the results-section: often the indicators in the control area are not mentioned, this is for dengue/Aedes very important, as we know that there is a high variability over seasons and years and often interferes with measurement of effectiveness. Best to always report on intervention and control area, or if there is no control area, to state it, so that the readers knows how to interpret the findings. This information need to be added to the table (effect in control area/or use relative effect/’reduction achieved’).

- Lines 184 and 187: contradiction: ‘there is an at least 40% decrease’ <-> ‘observed impact was low’: what do authors mean? Best to keep the facts in the results section and the interpretation in the discussions session. 

- Line 202: it reads as if in this study the LO were used to evaluate the intervention, but were not part of the control efforts – please clarify and if so, this study needs to be taken out

- Line 206-207: clarify this sentence – is this important information? It reads like a story and not as results of a research.

- Line 227 <-> line 232: in one sentence the effect is the same in the four sites, and further down authors state that the site with ‘integrated interventions’ is better than the other sites – please clarify

- Line 245-247: ? were the traps used as a control measure or as an evaluation tool?

- Chapter on ‘autocidal gravid ovitraps’ – reads too much as a story and not as a scientific article. For the reader it’s difficult to appraise scientific evidence from it.

- Line 271, line 280: reduction or relative reduction, taking into account the oscillation of parameters in control area?

- Insert something on comparability of areas pre-intervention

- Lines 372 – 376: opinion of authors or evidence from study?

- The title of the last column of the table 2: ‘reduction achieved’ – suggestion to make this ‘effectiveness measured’. I have also the impression that if there is no reduction found, that the authors don’t give a lot of information on effectiveness. This is biasing the readers with drawing too much attention to the studies with effects (and not reporting as good the studies without effect) – example study 32, study 18. Mind also to give the precise figures of the main indicator of each study.

- Information on public acceptability could be presented in a separate table

- Line 391: ‘no effect over entire period’ – but there was a significant effect in winter time – does this means that there was an adverse effect in summer? (as the average is zero effect, with a significant effect in one period)

- Lines 393-401: what is opinion of the authors and what is evidence?

- Lines 402 – 431: consider taking out of this review as it is not an effectiveness study, but efficacy studies in single (or 3) house(s). Also LO were tested in single houses and this was not included in this review.

**Conclusions**

-Are the conclusions supported by the data presented?

-Are the limitations of analysis clearly described?

-Do the authors discuss how these data can be helpful to advance our understanding of the topic under study?

-Is public health relevance addressed?

Reviewer #1: I strongly suggest to authors to revise the discussion section to expand specifically on the added value of the study on previous conducted studies. Additionally, the discussion would highly benefit from few bullet points of the main outcomes of the study. Authors should consider to include reference(s) on the main limitations of this study and any actions to mitigate them. 

Moreover, is important to discuss further why (or not) this method should adopted by stakeholders against Aedes invasive mosquitoes. 

The last comment (lines 570-576) is very important for mosquito control programs. Authors should consider connecting ovitraps with QC (there are 2 relevant references about evaluation of methoids from one of the authors; Michaelakis et al. 2021. International Journal of Environmental Research and Public Health, 18(7), p.3478 & Abramides et al. 2011. Transactions of the Royal Society of Tropical Medicine and Hygiene, 105(5), pp.281-288).

Reviewer #2: General conclusions are well supported by the data presented: “This review highlights a knowledge gap in demonstrating the effectiveness of mass trapping of mosquitoes in reducing viral transmission and disease. As a result, further large-scale Phase III cluster randomized controlled trials conducted in endemic areas and including epidemiological outcomes are needed to establish scientific evidence for the reduction of viral transmission risk by mass trapping targeting gravid and/or host-seeking female mosquitoes.”

Reviewer #3: - The discussion talks too much about evidence in vague terms, such as ‘potential evidence’, ‘could increase’, … 

- Line 448: report on the incidence/immunity to chikungunya prior to intervention in both areas, as this could influence effect

- Line 455: what is ‘aggressive density’ ?

- There is a lot of repetition in the discussion – it would improve if more concise and to the point 

- Line 482: what do authors mean with ‘exclusion areas to vector control’?

- Line 495: Social mobilization is normally added to an intervention strategy and is not an intervention on its own/it is a co-strategy to achieve mechanical elimination/control of breeding sites. Rephrase sentence.

- Line 507: the potential difference between urban and rural is an interesting focus, and was not sufficiently included in the review 

- Line 545 and following: why setting up a new repository if clintrials.gov already exists and contains several of the vector control studies. Perhaps an option making submission compulsory to the existing repositories? Line 555-558: too much speculation for a scientific paper?

**Editorial and Data Presentation Modifications?**

Reviewer #1: Title

It is not general “Aedes” but Aedes invasive species. 

Possible title could be: Current evidence on the effectiveness of mosquito mass-trapping interventions to control invasive Aedes populations and to reduce the transmission of their diseases: A systematic review

Abstract

Although authors consider “exclusion criteria” there are no exclusion criteria mentioned in the main text

A more general suggestion is that abstract would highly benefit from some more numerical data to support your main findings. 

Introduction

Authors should give a clear statement and information for invasive mosquitoes (Aedes) and their VBDs. 

Authors should provide more details on the management of these 2 Aedes. Indeed, it is too complicated, especially in urban areas. Why authors think that mass trapping is a potential method to reduce/control Aedes invasive populations?

Lines 60-68. Why do the authors make a specific reference on the case of France, while this is a global study? Maybe the study would benefit from a more global perspective and/or European. In Europe aegypti is not established so, discus the 2 species separately (e.g. for Ae. albopictus also check Bellini, et al 2020. Travel medicine and infectious disease, 35, p.101691.)

Lines 95-97. More references could be cited on the linkages of aedes and urban environments (especially systematic reviews). 

To support the objective of the study, authors should revise the introduction in a way that would highly benefit from further evidence on previous studies and how this study steps to them. In addition, the explanation of the scope of this review is very short and could be expanded to 3-4 bullet points.

Reviewer #3: (No Response)

PLOS authors have the option to publish the peer review history of their article (what does this mean?). If published, this will include your full peer review and any attached files.

Reviewer #1: No

Reviewer #2: No

Reviewer #3: No

**Summary and General Comments**

Reviewer #2: The authors present a systematic review following PRISMA guidelines of published evidence on the effectiveness of mosquito mass-trapping interventions to control Aedes populations and to reduce the transmission of Aedes-borne diseases. 

Authors discuss how data can be helpful to advance our understanding of the topic under study and its public health relevance. Limitations are mentioned. 

General conclusions are well supported by the data presented: “This review highlights a knowledge gap in demonstrating the effectiveness of mass trapping of mosquitoes in reducing viral transmission and disease. As a result, further large-scale Phase III cluster randomized controlled trials conducted in endemic areas and including epidemiological outcomes are needed to establish scientific evidence for the reduction of viral transmission risk by mass trapping targeting gravid and/or host-seeking female mosquitoes.”

However, some “conclusions” are incomprehensible. Particularly “All of these considerations point to the need for an international website on vector control trials. This website would be an online repository where teams conducting vector control trials could document the study design, type of intervention, geographic area targeted, objectives, endpoints, estimated duration, and sample size, and where contact information for the research team could be provided. This approach is used by most credible research teams working on human clinical trials and is becoming a regulatory requirement in many countries. Such a benchmark could be used by scientists around the world to assess other research being conducted elsewhere – where to fill in the gaps, where not to duplicate efforts, and where to create synergies. The rigor of these trials can be assessed by the quality of these protocols. Such labelling would also make it more difficult for commercial trapping systems to claim that their products have been properly tested and are effective if they have not. False claims of efficacy are a public health problem because people trust the wrong protection methods and are at greater risk of being infected.”

What is the proposal? An “new” international website on vector control trials? Is there any one now? Do the authors know VCAG?

If we don’t have the new website, what is being currently done is not credible? and not fulfilling regulatory requirements?

Labelling and claiming for commercial trapping systems is (and should) be addressed in each country; and indeed, should be honest and demonstrable.

Reviewer #3: The subject of this paper is interesting and important for decision making. However, it reads more as a story than as a systematic review. This could be improved upon by keeping the facts (results) in the results section and the interpretation in the discussions session, now there is a mix, which makes it hard to know what evidence is existing.

Abstract

- Specify geographical scope of review

- Findings are very poorly presented – insert more detail on range of effect that can be expected from such an intervention

- Conclusions: can be more to the point

Introduction:

- Not clear why there is a focus on France and the French territories in the introduction – would be better to open up to all endemic areas?
---

## [Decision Letter · Decision Letter 1]

24 Oct 2022

Dear Ms FITE,

Thank you very much for submitting your manuscript "Current evidences of the efficacy of mosquito mass-trapping interventions to reduce Aedes aegypti and Aedes albopictus populations and Aedes-borne transmission" for consideration at PLOS Neglected Tropical Diseases. As with all papers reviewed by the journal, your manuscript was reviewed by members of the editorial board and by several independent reviewers. The reviewers appreciated the attention to an important topic. Based on the reviews, we are likely to accept this manuscript for publication, providing that you modify the manuscript according to the review recommendations. 

Just a few minor revisions from one reviewer only

Sincerely,

Olaf Horstick, FFPH(UK)

Academic Editor

Karin Kirchgatter

Section Editor

Just a few minor revisions from one reviewer only

Reviewer's Responses to Questions

**Key Review Criteria Required for Acceptance?**

**Methods**

-Are the objectives of the study clearly articulated with a clear testable hypothesis stated?

-Is the study design appropriate to address the stated objectives?

-Is the population clearly described and appropriate for the hypothesis being tested?

-Is the sample size sufficient to ensure adequate power to address the hypothesis being tested?

-Were correct statistical analysis used to support conclusions?

-Are there concerns about ethical or regulatory requirements being met?

Reviewer #1: All the objectives and the design of the study are clearly explained.

Reviewer #2: (No Response)

Reviewer #3: o Please insert how quality/bias was evaluated in the papers selected - 

o Table 1 and Table 2: rephrase the definition of ‘comparator’ – what is now written in the box, is the description of study type (except between brackets)

o Table 2: study type: observational study is both an inclusion as an exclusion criterion

**Results**

-Does the analysis presented match the analysis plan?

-Are the results clearly and completely presented?

-Are the figures (Tables, Images) of sufficient quality for clarity?

Reviewer #1: All the results are clearly presented.

Reviewer #2: (No Response)

Reviewer #3: Effect/efficacy/effectiveness – the authors are mixing these concepts all over the text. Revise these concepts: for example - table 3 last column: use ‘effect’ in the title if there is a combination of efficacy and effectiveness studies (which is the case in this scoping review). Most of the studies are effectiveness studies, as they are applying tools in real life conditions, a few are efficacy studies as they are evaluating tools in few houses under very controlled circumstances. See also line 541, 532, 438, …

**Conclusions**

-Are the conclusions supported by the data presented?

-Are the limitations of analysis clearly described?

-Do the authors discuss how these data can be helpful to advance our understanding of the topic under study?

-Is public health relevance addressed?

Reviewer #1: All the conclusions are presented and public health relevance is clearly addressed.

Reviewer #2: (No Response)

Reviewer #3: See the remarks on the abstract and the synopsis - looks like conclusions stay very vague

lines 450-452 : what is an experimental bias?

**Editorial and Data Presentation Modifications?**

Reviewer #1: Accept

Reviewer #2: (No Response)

Reviewer #3: (No Response)

**Summary and General Comments**

Reviewer #1: Authors addressed all my concerns and suggestions. I strongly suggest them to highlight more their aspect for evaluation and monitoring of the mosquito control programs and support the last two paragraphs, in discussion, with relevant references.

Reviewer #2: The authors have followed and made the changes and suggestions of the reviewers in a satisfactory manner.

Reviewer #3: Most of the comments of first review were adequately addressed and paper improved consistently. However, here are still some remarks on abstract, synopsis and introduction

- Abstract: line 36 – ‘various studies support the efficacy….’. This is really too vague for an abstract. Alternatives: XX% of studies demonstrated a reduction of entomological indicators; or; another way could be to report on the quality of the studies included in the review – if you score them ‘low quality’, hence it is OK that you can’t say anything in the abstract. 

- Synopsis: ‘this review provides some evidences …’ – as in abstract, reads like authors are not convinced of the effect of these tools – to do with quality of papers included? – rephrase and be clear (as in abstract)

- Introduction:

o Line 71: rephrase ‘… with significant economic impacts.’

o Line 82: insert reference

o Line 84-88: no reference made to the systematic reviews done on larval control methods and adult control methods

o Line 95-96: … providing relevant entomological/epidemiological data on their efficacy’ – what is meant by this? Please add a reference

o Line 97: Authors rightfully refer to the review on traps of Johnson (reference 8) – but would be good to mention what this paper adds to the reference 8 (in introduction or discussion) – many of the conclusions are the same

o Lines 98-99: rephrase: ‘… complementary…complement …combined…’

PLOS authors have the option to publish the peer review history of their article (what does this mean?). If published, this will include your full peer review and any attached files.

Reviewer #1: No

Reviewer #2: No

Reviewer #3: No

Figure Files:

Data Requirements:

Reproducibility:

References

---

## [Editor Report · Decision Letter 2]

8 Feb 2023

Dear Ms FITE,

We are pleased to inform you that your manuscript 'Current evidences of the efficacy of mosquito mass-trapping interventions to reduce Aedes aegypti and Aedes albopictus populations and Aedes-borne transmission' has been provisionally accepted for publication in PLOS Neglected Tropical Diseases.

Best regards,

Karin Kirchgatter

Academic Editor

The comments of the reviewers were adequately addressed.

---

## [Editor Report · Acceptance letter]

1 Mar 2023

Dear Ms FITE,

We are delighted to inform you that your manuscript, "Current evidences of the efficacy of mosquito mass-trapping interventions to reduce *Aedes aegypti* and *Aedes albopictus* populations and *Aedes*-borne transmission ," has been formally accepted for publication in PLOS Neglected Tropical Diseases.

Best regards,

Shaden Kamhawi

co-Editor-in-Chief

Paul Brindley

co-Editor-in-Chief
